# DaXBench: Benchmarking Deformable Object Manipulation with Differentiable Physics

**Siwei Chen**[1†], **Yiqing Xu**[1†], **Cunjun Yu**[1†], **Linfeng Li**[1], **Xiao Ma**[2], **Zhongwen Xu**[2], **David Hsu**[1]

[1] National University of Singapore   [2] Sea AI Lab[‡]

## Abstract

Deformable object manipulation (DOM) is a long-standing challenge in robotics and has attracted significant interest recently. This paper presents DaXBench, a differentiable simulation framework for DOM. While existing work often focuses on a specific type of deformable objects, DaXBench supports fluid, rope, cloth ...; it provides a general-purpose benchmark to evaluate widely different DOM methods, including planning, imitation learning, and reinforcement learning. DaXBench combines recent advances in deformable object simulation with JAX, a high-performance computational framework. All DOM tasks in DaXBench are wrapped with the OpenAI Gym API for easy integration with DOM algorithms. We hope that DaXBench provides to the research community a comprehensive, standardized benchmark and a valuable tool to support the development and evaluation of new DOM methods. The code and video are available online[*].

## 1 Introduction

Deformable object manipulation (DOM) is a crucial area of research with broad applications, from household (Maitin-Shepard et al., 2010; Miller et al., 2011; Ma et al., 2022) to industrial settings (Miller et al., 2012; Zhu et al., 2022). To aid in algorithm development and prototyping, several DOM benchmarks (Lin et al., 2021; Huang et al., 2021) have been developed using deformable object simulators. However, the high dimensional state and action spaces remain a significant challenge to DOM.

Differentiable physics is a promising direction for developing control policies for deformable objects. It implements physical laws as differentiable computational graphs (Freeman et al., 2021; Hu et al., 2020), enabling the optimization of control policies with analytical gradients and therefore improving sample efficiency. Recent studies have shown that differentiable physics-based DOM methods can benefit greatly from this approach (Huang et al., 2021; Heiden et al., 2021; Xu et al., 2022; Chen et al., 2023). To facilitate fair comparison and further advancement of DOM techniques, it is important to develop a general-purpose simulation platform that accommodates these methods.

We present DaXBench, a differentiable simulation framework for deformable object manipulation (DOM). In contrast with current single-task benchmarks, DaXBench encompasses a diverse range of object types, including rope, cloth, liquid, and elasto-plastic materials. The platform includes tasks with varying levels of difficulty and well-defined reward functions, such as liquid pouring, cloth folding, and rope wiping. These tasks are designed to support high-level macro actions and low-level controls, enabling comprehensive evaluation of DOM algorithms with different action spaces.

We benchmark eight competitive DOM methods across different algorithmic paradigms, including sampling-based planning, reinforcement learning (RL), and imitation learning (IL). For planning methods, we consider model predictive control with the Cross Entropy Method (CEM-MPC) (Richards, 2005), differentiable model predictive control (Hu et al., 2020), and a combination of

---

[†]These authors contributed equally.

[‡]This work is partially completed at the SEA AI Lab.

[*]The link of the project is `https://github.com/AdaCompNUS/DaXBench`.

the two. For RL domains, we consider Proximal Policy Optimization (PPO) (Schulman et al., 2017) with non-differentiable dynamics, and Short-Horizon Actor-Critic (SHAC) (Xu et al., 2022) and Analytic Policy Gradients (APG) (Freeman et al., 2021) with analytical gradients. For IL, Transporter networks (Seita et al., 2021) with non-differentiable dynamics and Imitation Learning via Differentiable physics (ILD) (Chen et al., 2023) are compared. Our experiments compare algorithms with and without analytic gradients on each task, providing insights into the benefits and challenges of differentiable-physics-based DOM methods.

DaXBench provides a deformable object simulator, DaX, which combines recent advances in deformable object simulation algorithms (Xu et al., 2022; Chen et al., 2023) with the high-performance computational framework JAX (Bradbury et al., 2018). This integration allows for efficient auto-differentiation and parallelization across multiple accelerators, such as multiple GPUs. All task environments are wrapped with the OpenAI Gym API (Brockman et al., 2016), enabling seamless integration with DOM algorithms for fast and easy development. By providing a comprehensive and standardized simulation framework, we aim to facilitate algorithm development and advance the state of the art in DOM. Moreover, our experimental results show that the dynamics model in DaXBench enables direct sim-to-real transfer to a real robot for rope manipulation, indicating the potential applicability of our simulation platform to real-world problems.

## 2 RELATED WORKS

### 2.1 DEFORMABLE OBJECT SIMULATORS

Recent advancements in Deformable Object Manipulation (DOM) have been partially attributed to the emergence of DOM simulators in recent years. SoftGym (Lin et al., 2021) is the first DOM benchmark that models all liquid, fabric, and elastoplastic objects and introduces a wide range of standardized DOM tasks. ThreeDWorld (Gan et al., 2020) enables agents to physically interact with various objects in rich 3D environments, but it is non-differentiable and does not support methods based on differentiable physics.

On the other hand, several differentiable DOM simulators emerge, such as ChainQueen (Hu et al., 2019), Diff-PDE (Holl et al., 2020), PlasticineLab (Huang et al., 2021), DiSECt (Heiden et al., 2021), and DiffSim (Qiao et al., 2020). However, each simulator specializes in modeling a single type of deformable object and supports only a limited range of object-specific tasks. For example, PlasticineLab (Huang et al., 2021) and DiSECt (Heiden et al., 2021) model only the elasto-plastic objects, including tasks such as sculpting, rolling and cutting the deformable objects. Since the dynamics of each deformable object are vastly different, the physic engine specialized in modeling one type of deformable object cannot be easily extended to another. DaXBench bridges this gap by offering a general-purpose simulation framework that covers a wide range of deformable objects and the relevant tasks. We provide a fairground for comparing and developing all types of DOM methods, especially the differentiable ones.

### 2.2 DEFORMABLE OBJECT MANIPULATION ALGORITHMS

These challenges make it difficult to apply existing methods for rigid object manipulation directly to DOM. Methods designed for DOM must be able to handle a large number of degrees of freedom in the state space and the complexity of the dynamics. Despite these challenges, many interesting approaches have been proposed for DOM. Depending on the algorithmic paradigms, we categorize DOM methods into three groups: planning, Imitation Learning (IL), and Reinforcement Learning (RL). Our discussion below focuses on the general DOM methods for a standardized set of tasks. We refer to (Sanchez et al., 2018; Khalil & Payeur, 2010) for a more detailed survey on prior methods for robot manipulation of deformable objects.

**Reinforcement Learning** algorithms using differentiable physics such as SHAC (Xu et al., 2022) and APG (Freeman et al., 2021) are also explored. These methods use the ground-truth gradients on the dynamics to optimize the local actions directly, thereby bypassing the massive sampling for the policy gradient estimation.

**Imitation Learning** is another important learning paradigm for DOM. To overcome the "curse of history", the existing works such as (Seita et al., 2021; Sundaresan et al., 2020; Ganapathi et al.,

| Engine | Application | | | | Differen-tiability | Parallelization | |
| --- | --- | --- | --- | --- | --- | --- | --- |
| | Liquid | Fabric | Elasto-plastic | Mixture | | # of simulations per process | Multi-Device Training |
| PlasticineLab | ✗ | ✗ | ✓ | ✗ | ✓ | Single | ✗ |
| DiSECt | ✗ | ✗ | ✓ | ✗ | ✓ | Single | ✗ |
| DiffSim | ✗ | ✓ | ✗ | ✗ | ✓ | Single | ✗ |
| SoftGym | ✓ | ✓ | ✗ | ✗ | ✗ | Single | ✗ |
| DaXBench | ✓ | ✓ | ✓ | ✓ | ✓ | Multiple | ✓ |

Table 1: Comparison among the simulators for Deformable Object Manipulation.

2021) overcome the enormous state/action space by learning the low dimensional latent space and simplifying the primitive actions to only pick-and-place. ILD (Chen et al., 2023) is another line of IL method that utilizes the differentiable dynamics to match the entire trajectory all at once; therefore, it reduces the state coverage requirement by the prior works.

**Motion planning** for DOM has been explored in (Wi et al., 2022; Shen et al., 2022; Lippi et al., 2020; Nair & Finn, 2019; Yan et al., 2021; Ma et al., 2022). These methods overcome the enormous DoFs of the original state space by learning a low-dimensional latent state representation and a corresponding dynamic model for planning. The success of these planning methods critically depends on the dimensionality and quality of the learned latent space and the dynamic model, which in itself is challenging.

We believe that the development of a general-purpose simulation framework with a differentiable simulator can provide a platform for comparing existing DOM methods. Such a framework enables researchers to gain a better understanding of the progress and limitations of current DOM methods, which in turn can provide valuable insights for advancing DOM methods across all paradigms.

## 3 DAXBENCH: DEFORMABLE-AND-JAX BENCHMARK

DaXBench is a general-purpose differentiable DOM simulation platform that covers a wide range of deformable objects and manipulation tasks. To facilitate benchmarking of DOM methods from all paradigms, particularly differentiable methods, we developed our own differentiable simulator, Deformable-and-JAX (DaX), which is both highly efficient and parallelizable. In this section, we first provide a brief overview of our simulator, DaX, and then focus on introducing the benchmark tasks we have implemented for DaXBench.

### 3.1 DAX

DaX is a differentiable simulator designed to model various types of deformable object manipulation with high parallelization capabilities. A comparison with other existing simulators can be found in Table 1. DaX is implemented using JAX (Bradbury et al., 2018), which supports highly parallelized computation with excellent automatic differentiation directly optimized at the GPU kernel level. Moreover, DaX can be seamlessly integrated with the numerous learning algorithms implemented using JAX, allowing for the entire computation graph to be highly parallelized end-to-end.

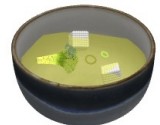

Figure 1: An example of a bowl of soup with tofu and vegetables. All materials including the soup are simulated by DaX using particles. Moreover, the entire simulation is fully differentiable and highly parallelizable.

**State Representations and Dynamic Models** DaX adopts different state representations and dynamic systems to model different types of deformable objects with distinct underlying deformation and dynamics. To model liquid and rope, which undergo significant deformation, DaX represents their states using particles and uses Moving Least Square Material Point Method (MLS-MPM) (Hu et al., 2018; Sulsky et al., 1994) to model their dynamics on the particle level. For cloth, which only undergoes limited deformation, DaX models its state using mesh and uses a less computationally expensive mass-spring system to model its dynamics. These modeling choices allow DaX to strike

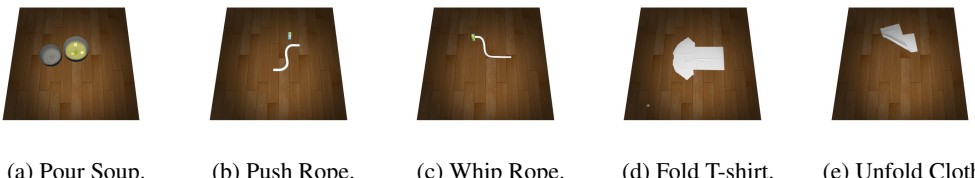

| (a) Pour Soup. | (b) Push Rope. | (c) Whip Rope. | (d) Fold T-shirt. | (e) Unfold Cloth. |

Figure 2: We illustrate different deformable object manipulation tasks in this work. A template is provided to readily expand this set to many other similar tasks.

a balance between modeling capacity and complexity. For more information on the object models used in DaX please refer to Appendix section A.1.

**Designed for Customization** DaX allows full customization of new tasks based on the existing framework. Objects with arbitrary shapes and different rigidness can be added to the simulator with ease. Moreover, primitive actions can also be customized. For a code example of how to customize a new environment in DaX, please refer to Appendix section A.2.

**Saving Memory** DaX uses two tricks to optimize the efficiency and memory consumption of the end-to-end gradient computation, namely lazy dynamic update and "checkpointing method" (Qiao et al., 2021; Chen et al., 2016; Griewank & Walther, 2000). To optimize the time and memory consumption of MPM for a single timestep, DaX lazily updates the dynamic of a small region affected by the manipulation at that step. Moreover, DaX only stores the values of a few states (i.e., checkpoints) during the forward pass. During the backward propagation, DaX re-computes segments of the forward values starting from every stored state in reverse order sequentially for the overall gradient. For more details on these techniques, please refer to Appendix section A.4.

### 3.2 BENCHMARK TASKS

DaXBench provides a comprehensive set of Deformable Object Manipulation tasks, including liquid pouring, rope wiping, cloth folding, and sculpting elastoplastic objects, as illustrated in Figure 2. These tasks are representative of the common deformable objects encountered in daily life. Additionally, we propose several new challenging long-horizon tasks that require finer-grained control, allowing for better benchmarking of the *scalability* of DOM methods. The horizon of each task is determined by how many steps a human expert takes to finish the task. DaXBench provides a standardized platform for comparing existing methods and future work on DOM. In this section, we provide detailed information on the tasks supported by DaXBench.

**Long Horizon Tasks** The time complexity for the search-based planning method grows exponentially with the planning horizon, known as the "curse of history"; this is exacerbated by the large state space of deformable objects. However, the existing DOM benchmark tasks typically have relatively short horizons. For example, the cloth folding and unfolding tasks in SoftGym consist of only 1-3 steps. DaXBench addresses this limitation by providing tasks with longer horizons, enabling better evaluation of the scalability of search-based planning methods.

**Tasks without Macro-Actions** A macro-action can be seen as executing raw actions for multiple steps. For example, we can define a *pick-and-place* or *push* macro-action as a 6-dimensional real vector $(x, y, z, x', y', z')$ representing the start $(x, y, z)$ and end $(x', y', z')$ positions of the gripper, and the raw actions are Cartesian velocities of the gripper. Macro-actions reduce the dimension of the action space and the effective horizon, and hence make scaling easier for the DOM methods. However, macro-actions might be too coarse for some tasks. For example, when whipping a rope to hit a target location, the momentum needs to be adjusted at a relatively high frequency throughout the execution, making macro-actions unsuitable. DaXBench includes tasks that do not have suitable macro-actions to provide a more comprehensive evaluation of DOM methods.

#### 3.2.1 LIQUID MANIPULATION

**Pour-Water** Pour a bowl of water into the target bowl as quickly as possible. The agent directly controls the velocity and rotation (in Cartesian space) of the gripper that holds the bowl. In particular,

the action space of the agent is a 4-dimensional vector $a = (v_x, v_y, v_z, w)$, for which $(v_x, v_y, v_z)$ represents the linear velocity of the gripper, and $w$ denotes the angular velocity around the wrist.

**Pour-Soup** Pour a bowl of soup with various solid ingredients into the target bowl as fast as possible. The actions are the same as those of Pour-Water. This task is more challenging than Pour-Water due to the additional interactions between the liquid and the solid ingredients. In particular, the agent needs to promptly react when a solid ingredient is transferred, so as to 1) adjust for the sudden decrease in load on the gripper and 2) control the momentum of the ingredient when hitting the soup surface to minimize spilling.

Neither tasks have suitable high-level macro-actions and the task horizons are 100 steps.

### 3.2.2   ROPE MANIPULATION

**Push-Rope** Push the rope to the pre-specified configuration as quickly as possible. We randomly initialize the configuration of the rope. The agent uses a single gripper to execute the *push* macro-actions. The horizon for this task is 6 steps.

**Whip-Rope** Whip the rope into a target configuration. The agent holds one end of the rope and controls the gripper's velocity to whip the rope. In particular, the action space of this task is a 3-dimensional vector $a = (v_x, v_y, v_z)$ that denotes the velocity of the gripper in the Cartesian space. This task cannot be completed by the abstract macro-actions since the momentum and the deformation of the rope has to be adjusted at a relatively high frequency. The task horizon is 70 steps.

### 3.2.3   CLOTH MANIPULATION

**Fold-Cloth** Fold a piece of flattened cloth and move it to a target location. This task has two variants: the easy version requires the agent to execute 1 fold and the difficult version 3 folds. For both variants, the task begins with the cloth lying flat on the table at an arbitrary position. The agent executes *pick-and-place* macro-actions. The task horizon is 3 and 4 for the easy and difficult versions respectively.

**Fold-T-shirt** Fold a T-shirt to a target location. The task begins with the T-shirt lying flat on the table at an arbitrary position. The agent executes *pick-and-place* macro-actions. The task horizon is 4.

**Unfold-Cloth** Flatten a piece of folded cloth to a target location as fast as possible. This task also has two variants that differ in the initial configurations of the folded cloth: the cloth in the easy version has been folded once and that in the difficult version has been folded three times. This task has the same action space as the Fold-Cloth task. The task horizon is 10 steps for both variants.

All of these tasks share similar APIs to the OpenAI Gym's standard environments. We provide a general template to define tasks in our simulator. This not only better organizes the existing task environments but also allows the users to customize the existing task environments to better suit their research requirements. The variables and constants defined in the template are easily interpretable quantities that correspond to real physical semantics. This allows the users to better understand the environment, which can also help with the development and debugging process.

## 4   EXPERIMENTS

In this section, we aim to answer the following two questions: 1) Are the existing representative methods capable of solving the tasks in DaXBench? 2) Within each paradigm, will the differentiable simulator help in completing the tasks?

To investigate these two questions, we benchmark eight representative methods of deformable object manipulation, covering planning, imitation learning, and reinforcement learning. Each paradigm has different assumptions and requirements for input knowledge. We summarize the baseline methods and the relevant information in Table 2.

| Paradigms | Reinforcement Learning | | | Imitation Learning | | Planning | | |
|---|---|---|---|---|---|---|---|---|
| Methods | PPO | SHAC | APG | Transporter | ILD | CEM-MPC | diff-MPC | diff-CEM-MPC |
| # expert demos | — | — | — | 10 | 1 | — | — | — |
| Reward function | ✓ | ✓ | ✓ | — | — | ✓ | ✓ | ✓ |
| Differentiability | ✗ | ✓ | ✓ | ✗ | ✓ | ✗ | ✓ | ✓ |

Table 2: **Methods on Deformable Object Manipulation with respect to different setups.**

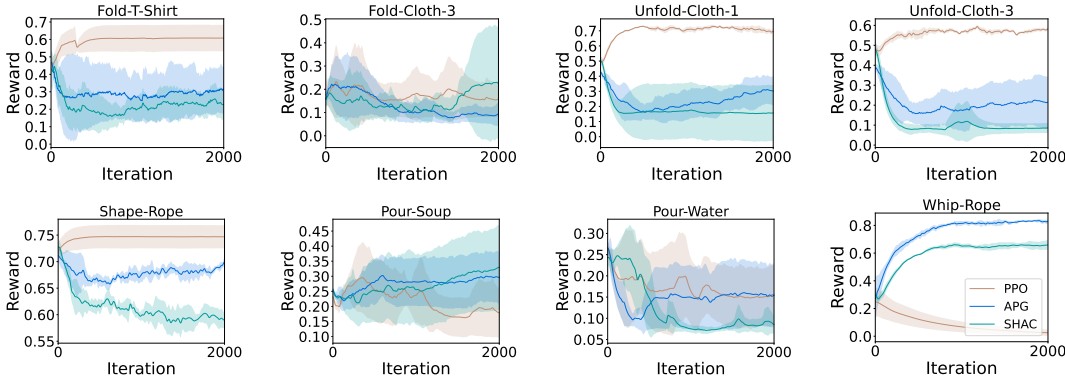

Figure 3: **Learning curves of three benchmarked RL algorithms: PPO, APG, and SHAC.** The y-axis shows the episode reward, scaled from 0 to 1, with larger values being closer to the goal state. The x-axis is the number of training iterations. All methods are trained in the same number of batched environments. We report the mean and variance of the performance for 5 seeds and each seed has 20 rollouts with different initializations. We refer the readers to Table 4 for the numerical results in the appendix.

## 4.1 EXPERIMENTAL SETUP

**Reward function** For each task, the goal is specified by the desired final positions of the set of particles, $g$, representing the deformable object. In this setup, the reward can be intuitively defined by how well the current object particles match with those of the goal. Hence, we define the ground-truth reward as $r_{gt}(s, a) = \exp(-\lambda D(s', g))$, where $s'$ is the next state resulted from $a$ at the current state $s$, and $D(s, g)$ is a non-negative distance measure between the positions of the object's particles $s$ and those of the goal $g$.

However, using the ground-truth reward alone is inefficient for learning. The robot's end-effector can only change the current object pose by contacting it. Therefore, to most efficiently manipulate the object to the desired configuration, we add an auxiliary reward to encourage the robot to contact the object. We define the auxiliary reward as $r_{aux}(s, a) = \exp(-L_2(s, a))$, where $L_2(s, a)$ measures the $L2$ distance between the end-effector and the object. During training, we use the sum of the ground-truth and auxiliary reward functions $r_{gt} + r_{aux}$, and during evaluation, we only use the ground-truth reward $r_{gt}$.

## 4.2 REINFORCEMENT LEARNING

**Methods.** We benchmark PPO (Schulman et al., 2017), SHAC (Xu et al., 2022), and APG (Freeman et al., 2021) as the representative RL methods. PPO is a widely used model-free non-differentiable RL method with competitive performance, while SHAC and APG are the two latest RL methods that utilize the differentiable simulator. The computation graph of SHAC is illustrated in Figure 4a. By comparing the performance of the DaXBench tasks across these three methods, we aim to further study the benefits and limitations of applying the differentiable physics to RL.

**Overall Performance.** The learning curves of RL methods are shown in Figure 3. The performance difference between PPO and the differentiable RL methods shows opposite trends in the high-level macro-action-based and local-level control tasks. To our surprise, for most of the short-horizon

| Task Type | Task | Imitation Learning | | | Planning | | |
|---|---|---|---|---|---|---|---|
| | | Transporter | ILD | Expert | CEM-MPC | Diff-MPC | Diff-CEM-MPC |
| High-level | Fold-Cloth-1 | 0.19±0.04 | 0.76±0.04 | 0.91±0.00 | 0.74±0.07 | 0.29±0.07 | 0.80±0.04 |
| | Fold-Cloth-3 | 0.40±0.05 | 0.82±0.05 | 0.89±0.00 | 0.62±0.08 | 0.14±0.02 | 0.66±0.07 |
| | Fold-T-shirt | 0.46±0.02 | 0.59±0.11 | 0.85±0.00 | 0.58±0.04 | 0.29±0.06 | 0.66±0.01 |
| | Unfold-Cloth-1 | 0.48±0.01 | 0.64±0.03 | 0.87±0.00 | 0.57±0.04 | 0.30±0.05 | 0.67±0.02 |
| | Unfold-Cloth-3 | 0.30±0.00 | 0.52±0.02 | 0.87±0.00 | 0.57±0.01 | 0.26±0.04 | 0.58±0.01 |
| | Push-Rope | 0.70±0.00 | 0.76±0.02 | 0.93±0.00 | 0.71±0.01 | 0.70±0.01 | 0.71±0.01 |
| Low-level | Whip-Rope | — | 0.70±0.06 | 1.00±0.00 | 0.34±0.01 | 0.37±0.00 | 0.33±0.01 |
| | Pour-Water | — | 0.32±0.03 | 0.91±0.06 | 0.58±0.01 | 0.30±0.00 | 0.58±0.01 |
| | Pour-Soup | — | 0.42±0.12 | 0.85±0.00 | 0.56±0.01 | 0.44±0.00 | 0.55±0.02 |

Table 3: **Task performance for the planning and imitation learning methods.** We report the mean and standard error for the policy/control sequences evaluated under 20 seeds.

macro-action-based tasks, PPO performs consistently better than the differentiable RL methods. This seems to contradict the experiment results of SHAC (Xu et al., 2022), which shows stronger performance than PPO in its own simulator on a wide range of context-rich MuJoCo tasks. However, for long-horizon tasks with low-level controls, the differentiable RL methods, especially APG, outperform PPO by a large margin.

**Main Challenge for the differentiable-physics-based RL methods: Exploration.** We argue that, on high-level macro-action-based tasks, the main limiting factor of the performance of differentiable-physics-based RL is the lack of exploration during training. The need to balance this trade-off is exaggerated by the enormous DoFs and the complex dynamics unique to the DOM tasks. Especially for the high-level macro-action-based tasks, their non-smooth/discontinuous/non-convex optimization landscape necessitates using a good exploration strategy. However, the existing differentiable-physics-based RL methods rely exclusively on the simulation gradient to optimize the policy. They are not entropy regularized; hence, they may quickly collapse to the local optimal. Without explicit exploration, the differentiable-physics-based RL methods fail to reach the near-optimal state using the simulation gradients alone. Taking Fold-Cloth-3 as an example, if the randomly initialized policy never touches the cloth, the local simulation gradients cannot guide the agent to touch the cloth since the rewards and, consequently, the gradients for all not-in-contact actions are zero.

**Differentiable-physics-based RL are sensitive to the Optimization Landscape.** Differentiable-physics-based RL can largely improve the performance of some low-level control tasks if the optimization landscape is smooth and convex. Taking Whip-Rope as an example, the optimal low-level control sequence is a smooth motion trajectory that lies on a relatively smooth and well-behaved plane for the gradient-based methods. This explains why APG outperforms the non-differentiable-physics-based PPO by a large margin.

## 4.3 IMITATION LEARNING

**Methods.** We benchmark Transporter (Seita et al., 2021) and ILD (Chen et al., 2023) as the two representative Imitation Learning DOM methods. Transporter is a popular DOM method that is well-known for its simple implementation and competitive performance. We use particles as input to evaluate the performance of Transporter in our experiment for a fair comparison. The original implementation of Transporter uses RGBD as input. We refer the readers to the Appendix section D for the performance of the Transporter with RGB channels. ILD is the latest differentiable-physics-based IL method; it utilizes the simulation gradients to reduce the required number of expert demonstrations and to improve performance. Note that since Transporter abstracts its action space to *pick-and-place* actions to simplify the learning, it cannot be extended to the low-level control tasks. Our comparison of the IL methods will first focus on the high-level macro-action-based tasks, then we will analyze the performance of ILD with the expert.

**Overall performance** For the high-level macro-action-based tasks, ILD outperforms Transporter by a large margin for all tasks. For example, ILD outperforms Transporter in Fold-Cloth-1, Fold-Cloth-3, and Unfold-Cloth-1. We highlight that this significant performance gain is achieved by using a much smaller number of expert demonstrations. We attribute this improvement to the additional information provided by the simulation gradients, where now ILD can reason over the ground-truth

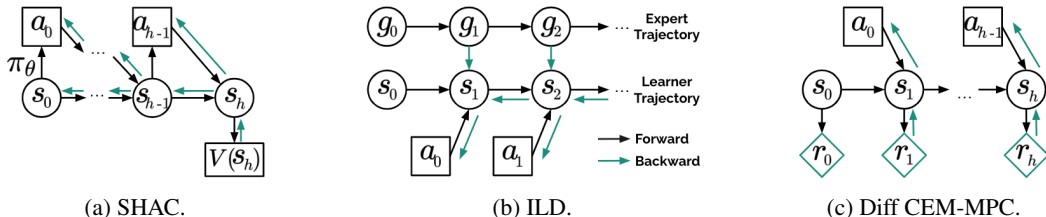

(a) SHAC.           (b) ILD.          (c) Diff CEM-MPC.

Figure 4: **Computation graphs for the differentiable algorithms.** We use SHAC, ILD, and Diff CEM-MPC to represent the differentiable algorithms for planning, RL, and IL respectively. The black/green arrows correspond to the forward/backward propagation respectively.

dynamics to bring the trajectory closer to the expert, at least locally. As illustrated in Figure 4, the gradients used to optimize each action comes from the globally optimal expert demonstrations 1) at the current step and 2) in all the steps after. In contrast, the gradients used in SHAC only count the reward at the current step. The differentiable RL faces the challenge of exploding/vanishing gradients and lack of exploration. Differentiable IL alleviates these problems by having step-wise, globally optimal guidance from expert demonstrations.

**Challenge for ILD: Unbalanced State Representations.** ILD does not perform well on tasks like Pour-Water and Pour-Soup with unbalanced state representations, for example, $1000 \times 3$ features of particles and 6 gripper states. The learned signals from the particles overwhelm the signals from the gripper states. Therefore, ILD cannot learn useful strategies with unbalanced learning signals. Softgym (Lin et al., 2021) also concurs with the results that policies with reduced state representation perform better than those with the full state. A more representative representation of the states needs to be considered and we leave it for future study.

In conclusion, differentiable methods can optimize the policies efficiently if the optimization landscape is well-behaved. For tasks that require tremendous explorations or with sparse rewards, gradient-based methods suffer severely from the local minima and gradient instability issues.

### 4.4 PLANNING

**Methods.** We benchmark the classic CEM-MPC (Richards, 2005) as a representative random shooting method that uses a gradient-free method to optimize the action sequences for the highest final reward. To study the effect of using differentiable physics, we implement two versions of differentiable MPC. The basic computation graph for both is illustrated in Figure 4c. In particular, both differentiable MPC baselines maximize the immediate reward of each action using differentiable physics. The difference lies in the initial action sequences: diff-MPC is initialized with a random action sequence, while diff-CEM-MPC with an action sequence optimized by CEM-MPC.

**Overall Performance.** The diff-CEM-MPC performs consistently better than or on par with CEM-MPC, while Diff-MPC fails to plan any reasonable control sequences for most of the tasks.

**Simulation Gradients as Additional Optimization Signals.** The vastly different performance between the two versions of the differentiable baselines brings us insights into how differentiable physics helps in planning. Simulation gradients provide additional signals in the direction to optimize the actions, thereby reducing the sample complexity. However, similar to any gradient-based optimization methods, differentiable planning methods are sensitive to the non-convex/non-smooth optimization landscape and face a major limitation of being stuck at the local optimal solution. DOM's enormous DoFs of DOM and complex dynamics exacerbate these challenges for the differentiable DOM planning methods.

**Differentiable Planning needs Good Initialization.** We attribute the poor performance of Diff-MPC to its random control sequence initialization. In particular, this control sequence may significantly deviate from the optimal one over the non-smooth/non-convex landscape, such that the simulation gradients alone cannot bring the control sequence back to the optimal. In contrast, Diff-CEM-MPC is initialized with the optimized control sequence by CEM-MPC, which relies on iterative sampling followed by elite selection to escape the local optimality. Hence, the control sequence fed into the differentiable step is already sufficiently close to the globally optimal solution. The

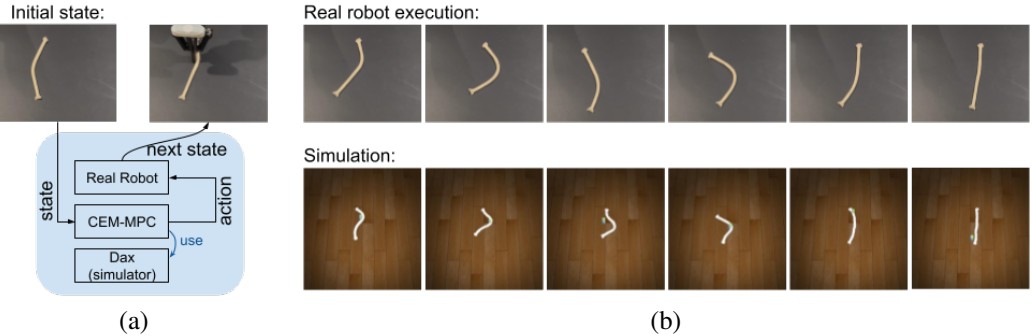

(a)  (b)

Figure 5: **We deploy CEM-MPC in a Push-Rope task to straighten the rope in 6 steps on a real robot**. (a) The system structure. CEM-MPC uses our DaX simulator as the predictive model. Given the current state estimated from a point cloud image, CEM-MPC plans the best action to be executed by the robot. (b) The states after each of the 6 pushes. We compare the dynamics in reality and in simulation. Top: state trajectory in a real robot experiment. Bottom: the next state predicted by DaX given the estimated current state and the corresponding planned action.

simulation gradients help diff-CEM-MPC to optimize the action sequences further locally, which explains the performance gain.

To summarize, the differentiable simulator can further optimize the near-optimal control sequences locally with much low sample complexity. Yet, its performance critically relies on a good initialization and a locally smooth optimization landscape.

### 4.5 SIM-2-REAL GAP

Granted that the physical engine of DaX is built upon a state-of-the-art simulation method, the fidelity and correctness still need to be tested. To testify to the correctness of our simulated dynamics to real-world physics, we carry out a real robot experiment on a Kinova Gen 3 robot. We deploy CEM-MPC with DaX as the predictor model to a Push-Rope task, as shown in Figure 5. Our study finds that the resultant state trajectories are similar in simulation and in reality. This validates the fidelity of the tasks implemented based on our engine, and the performance of the DOM methods in our simulator provides hints about their performance on the real robot. We refer the reader to the appendix for details. In addition, the differentiability of DaX enables system identification that can reduce the sim-2-real gap (Le Lidec et al., 2021; Murthy et al., 2021). We can get an error term by comparing the observations in simulation and in reality; by differentiating this error with respect to the simulator parameters, we get gradients for parameter tuning.

## 5 CONCLUSION

We introduced DaXBench, a general-purpose differentiable simulation framework for various deformable object manipulation tasks. To our knowledge, DaXBench is the first platform that simulates liquid, rope, fabric, and elastoplastic materials while being differentiable and highly parallelizable. In addition to common DOM tasks, we also proposed several novel tasks, including WhipRope and PourSoup, to study the performance of DOM methods on long-horizon tasks with low-level controls. Our task coverage provides a comprehensive platform to compare existing DOM methods from all paradigms, and our empirical study sheds light on the benefits and limitations of differentiable-physics-based methods in DOM. We believe that DaXBenchcan assist in the algorithmic development of DOM methods, particularly those based on differentiable physics, across all paradigms.

ACKNOWLEDGEMENTS

This research is supported by the National Research Foundation, Singapore under its AI Singapore Programme (AISG Award No: AISG2-PhD-2021-08-015T, AISG2-PhD-2022-01-036T, and AISG2-PhD-2021-08-014) and A*STAR under the National Robotics Program (Grant No. 192 25 00054).

ETHICS STATEMENT

DaXBench provides a benchmark for deformable object manipulation (DOM). We believe that DaXBench and its simulator DaX have no potential negative societal impacts. In addition, our work is on benchmarking the DOM methods in simulator, so we do not collect any data or conduct experiments with human subjects. In summary, we have read the ICLR Code of Ethics and ensured that our work conforms to them.

REPRODUCIBILITY STATEMENT

In this paper, imitation learning and planning experiments are averaged over 20 random rollouts and RL approaches use 100 rollouts. We have included our source code as an easy-to-install package in the supplementary material.

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

## A  IMPLEMENTATION DETAILS

### A.1  OBJECTS

DaXBench models a variety of deformable objects of different dimensionalities, ranging from 0-dimensional liquid, 1-dimensional rope, to 2-dimensional cloth. The underlying deformation and the dynamics vary distinctly for each dimension. In general, the lower the dimension is, the higher the degrees of freedom (DoFs) of the system, thus the more complex modeling that system becomes. We use two different representations to model the deformable objects to balance the modeling capacity with its complexity: water and rope are modeled using Material Point Method (MPM), while the cloth is modeled using a mass-spring system.

**Modelling Water and Rope using MPM** Material Point Method (MPM) demonstrates strong modeling capacity and flexibility. MPM models the deformable objects as a set of particles; each particle is represented by its position and some properties such as mass. The dynamics and the deformation of the object are computed by simulating the interactions among all individual particles, conforming to some pre-defined physics rules. This Particle-level representation and the dynamics simulation allow MPM to model the object under severe deformation. We, therefore, choose to use MPM to model water and rope that can undergo severe deformation, for they can exploit the model capacity of MPM to its full potential.

**Modelling Cloth using Mass-Spring System** Cloth only undergoes limited deformation as it has to conform to the canonical shape of the initial cloth through manipulation. This limited deformable does not require the strong modeling capacity of MPM, which is computationally expensive. Therefore, rather than modeling the deformation on the Particle-level, we choose to model the cloth using the mass-spring system, which is a simpler model that uses the physical constraints as a prior.

### A.2  CODE EXAMPLE

```
import numpy as np
from core.envs.registration import env_functions

env = env_functions["fold_cloth3"](batch_size=32, seed=1)
obs, state = env.reset(env.simulator.key_global)

actions = np.random.uniform(0, 1, (env.batch_size, env.action_size))
obs, reward, done, info = env.step_diff(actions, state)
```
Listing 1: Example of Running Cloth Environment

```
import jax
import numpy as np
from core.envs.registration import env_functions

env = env_functions["fold_cloth3"](batch_size=32, aux_reward=True)
obs, state = env.reset(env.simulator.key_global)
actions = np.random.uniform(0, 1, (env.batch_size, env.action_size))

@jax.jit
@jax.grad
def compute_grad(actions, state):
    obs, reward, done, info = env.step_diff(actions, state)
    return -reward.sum()

print("action gradients", compute_grad(actions, state))
```
Listing 2: Example of Computing Action Gradients via DaX

### A.3  JAX TO PARALLELIZE

One major limitation of the existing simulators is the lack of support for highly paralleled sampling. The key challenges for Deformable Object Manipulation are the complex governing equations and

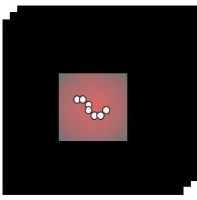
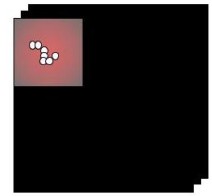
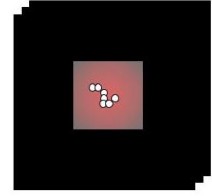

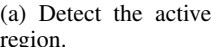

(a) Detect the active region.

(b) Update the deformation.

(c) Shift back the update active region.

Figure 6: We demonstrate how to use the active region to reduce the computation complexity for the simulation. Taking rope for example, the original environment, denoted as the black background, is of size $128 \times 128 \times 128$, while the active region, where the region that will be actively changed due to the current deformation highlighted in red, is of size $32 \times 6 \times 32$, as illustrated in (a). To avoid updating all the particles in the environment, in ( b), we shift the active region to the top corner, the *origin* of the environment, and update the dynamic and observation. Finally, in ( c), we shift the updated active region back to align its original coordinates.

the enormous state space. The complex governing equations invalidate most of the analytical approaches since solving the complex governing equations is intractable. On the other hand, the success of the sampling-based methods hinges on the reasonable coverage of the state space. The lack of support for highly paralleled sampling to better cover the state space is a major impediment to the works on Deformable Object Manipulation. DaXBench fills in this gap by implementing its simulator using JAX, which supports highly paralleled sampling with great automatic differentiation. Using our implementation, we can finish an iteration (both forward and backward pass) for 128 rollouts with 80 timesteps on a server with 4 2080-Ti GPUs in 3 seconds.

### A.4 SAVING MEMORY

**Lazy Dynamic Update** DaXBench optimizes the time and memory consumption of MPM by only updating the dynamic of a small region affected by the manipulation at each step. MPM models and updates the state and dynamics on the Particle-level; its time and memory consumption is proportional to the size of the environment, which can be arbitrarily large. Our insight is that, though the deformable object can potentially interact with the entire environment, only a small region will be affected at each time. This inspires DaXBench to optimize the time and memory consumption of running MPM by exclusively considering the *active region* during the interactions. Using this trick, DaXBench effectively reduces the computation time by two orders of magnitude on average. Taking the rope for example, the original environment is of size $128 \times 128 \times 128$. However, a grid of size $32 \times 6 \times 32$ is sufficient to cover the region of interaction between the environment and the rope. Therefore, we first detect the *active region* of size $32 \times 6 \times 32$, then we perform a linear transformation to shift the *active region* to the *origin*, perform the update due to deformation and interaction, finally we shift the updated *active region* to align with its original coordinates.

**Checkpointing method** DaXBench only stores the values of a few states (i.e. checkpoints), and re-evaluates the sub-step gradients online during the back-propagation to reduce the memory consumption for long horizon problems. The checkpointing method has been proven useful in saving the memory consumption for differentiable simulations in many prior works (Qiao et al., 2021; Chen et al., 2016; Griewank & Walther, 2000). To make the simulator end-to-end differentiable, every value in the neural network that is computed during the forward pass has to be saved, as they are needed to compute the gradient during the backward pass. The computation complexity scales up quickly w.r.t. the length of the task horizon. This is a significant problem for our simulator for high DoFs deformable object manipulation, as the memory required for each timestep's update is enormous compared to its rigid body counterpart. To overcome the large memory consumption issue, DaXBench trade-offs the computation complexity slightly for large memory consumption improvement: DaXBench only stores the values of a few states (i.e. checkpoints) at the forward pass; during the backward propagation, DaXBench re-compute segments of the forward values starting from every stored state in the reverse order sequentially for the overall gradient. With this trick, DaXBench

| Task Type | Task | PPO | APG | SHAC |
|---|---|---|---|---|
| High-level | Fold Cloth 1 | **0.40±0.13** | 0.36±0.06 | 0.34±0.07 |
| | Fold Cloth 3 | 0.21±0.11 | 0.19±0.09 | **0.22±0.22** |
| | Fold T-shirt | **0.61±0.08** | 0.40±0.05 | 0.44±0.09 |
| | Unfold Cloth 1 | **0.72±0.01** | 0.42±0.02 | 0.50±0.03 |
| | Unfold Cloth 3 | **0.56±0.00** | 0.39±0.02 | 0.48±0.03 |
| | Rope Push | **0.75±0.02** | 0.72±0.02 | **0.75±0.01** |
| Low-level | Wipe Rope | 0.25±0.10 | **0.83±0.01** | 0.66±0.03 |
| | Pour Water | 0.27±0.02 | **0.27±0.02** | **0.28±0.00** |
| | Pour Soup | 0.27±0.08 | 0.27±0.00 | 0.32±0.13 |

Table 4: Reinforcement Learning Methods Task Results. We report the mean and standard error over 20 random seeds for each entry.

now doubles the computation complexity, but saves an arbitrary amount of memory, depending on the intervals of the states saved.

### A.5 DISCONTINUOUS GRADIENT

As an end-to-end differentiable simulator, the effectiveness of DaXBench rests on the accuracy of the gradient computed. We found that for deformable object manipulation, the action not in contact with the object will cause the gradient to be discontinuous, which causes the estimate of the gradient after this action to suffer from high variance and poor accuracy in the backward propagation. To mitigate this problem, for every action input by the agent or the human user, DaXBench will perform local adjustment such that the action will always make the gripper in contact with the deformable objects. This ensures that our gradient is end-to-end continuous therefore improving the accuracy of the estimated gradient.

## B SIM-2-REAL GAP

To verify that the dynamics of our simulated tasks have high fidelity and a small Sim2Real gap, we carry out a real robot experiment. We deploy CEM-MPC to the PushRope task on a Kinova Gen3 robot. CEM-MPC uses our DaX simulator as the predictive model. Given a state, CEM-MPC plans the next best *push* action $(x, y, z, x', y', z')$. The state is estimated from a point cloud image, and the Cartesian space *push* action is transformed to the joint space trajectory via an inverse kinematics module. Note we are not verifying the sim2real gap caused by the inaccuracy of state estimation or kinematics; we are interested in whether the dynamics of DaX can correctly guide the CEM-MPC to succeed in the task. The experiment is included in our supplementary video.

## C RL ALGORITHMS NUMERICAL EXPERIMENT RESULTS

We report the numerical experiment performance for the RL algorithms in Table 4. This is to supplement the learning curve in the main text. These numerical results report the final performance of the learning curves in Figure 3.

| Task Type | Task | Imitation Learning | | | |
|---|---|---|---|---|---|
| | | Transporter | Transporter-RGB | ILD | Expert |
| High-level | Fold-Cloth-1 | 0.19±0.04 | 0.86±0.07 | 0.76±0.04 | 0.91±0.00 |
| | Fold-Cloth-3 | 0.40±0.05 | 0.56±0.25 | 0.82±0.05 | 0.89±0.00 |
| | Fold-T-shirt | 0.46±0.02 | 0.83±0.05 | 0.59±0.11 | 0.85±0.00 |
| | Unfold-Cloth-1 | 0.48±0.01 | 0.76±0.06 | 0.64±0.03 | 0.87±0.00 |
| | Unfold-Cloth-3 | 0.30±0.00 | 0.70±0.06 | 0.52±0.02 | 0.87±0.00 |
| | Push-Rope | 0.70±0.00 | 0.86±0.09 | 0.76±0.02 | 0.93±0.00 |
| Low-level | Whip-Rope | — | — | 0.70±0.06 | 1.00±0.00 |
| | Pour-Water | — | — | 0.32±0.03 | 0.91±0.06 |
| | Pour-Soup | — | — | 0.42±0.12 | 0.85±0.00 |

Table 5: **Task performance for the planning and imitation learning methods.** We report the mean and standard error for the policy/control sequences evaluated under 20 seeds.

## D TRANSPORTER WITH IMAGE AS INPUT

We reported the performance of two different versions of Transporter in Table 5: Transporter, which uses 3D particles and projects them onto an image to form a height map; and Transporter-RGB, which adds RGB channels and projects the particles onto the image plane, giving it white color with a black background. We have found that an explicit process of rendering particles into RGB images significantly enhances the performance of Transporter.

