# OpenReview forum: "DaxBench: Benchmarking Deformable Object Manipulation with Differentiable Physics"
_ICLR.cc/2023/Conference — ICLR 2023 notable top 5%_

### Official Review · Reviewer_5D3R · 2022-10-22

**Confidence:** 4
**Clarity, Quality, Novelty And Reproducibility:** Great on all fronts.
**Correctness:** 3
**Technical Novelty And Significance:** 3
**Empirical Novelty And Significance:** 3
**Recommendation:** 8

**Strength And Weaknesses:**

Strength:
- A very respectable implementation efforts was put into this work. I think this simulator will be of interest to many in the community.
- A large range of learning algorithms are studied. Offering some interesting insights.


Weakness:
- Some of the more recent DOM works are not referenced to, all of these works offer insights / simulator for DOM:
* ThreeDWorld: A Platform for Interactive Multi-Modal Physical Simulation, Gan et al, Neurips 2021
* ACID: Action-Conditional Implicit Visual Dynamics for Deformable Object Manipulation Bokui Shen, Zhenyu Jiang, Christopher Choy, Leonidas J. Guibas, Silvio Savarese, Anima Anandkumar, Yuke Zhu, RSS, 2022
* VIRDO: Visio-tactile Implicit Representations of Deformable Objects., Youngsun Wi, Pete Florence, Andy Zeng, Nima Fazeli. ICRA 2022.
- I'm not confident about the sim2real part. Saying the simulator has sim2real ability is a pretty big claim. However, only a demo video is provided without any systematic evaluation.

**Summary Of The Paper:**

The paper introduce DaXBench, which is a very strong differentiable simulator for various DOM tasks. The authors examined various algorithms for the given tasks.

**Summary Of The Review:**

Very nice paper. I think it's a strong paper and definitely worth acceptance. I wish the authors can address the weakness i mentioned above.

---

> ### Author Response · Authors · 2022-11-14
> **Authors Response**
>
> Thank you for recognizing the contribution of DaxBench! We would like to first respond to the missing related works and then explain our Sim2Real experiments. We’ve edited the manuscript to add the missing literature on DOM. Thank you for pointing this out and helping us in making our work better!
>
> &nbsp;
>
> > **Q1**:Some of the more recent DOM works are not referenced to, all of these works offer insights / simulator for DOM:
>
> - ThreeDWorld: A Platform for Interactive Multi-Modal Physical Simulation, Gan et al, Neurips 2021
> - ACID: Action-Conditional Implicit Visual Dynamics for Deformable Object Manipulation Bokui Shen, Zhenyu Jiang, Christopher Choy Leonidas J. Guibas, Silvio Savarese, Anima Anandkumar, Yuke Zhu, RSS, 2022
> - VIRDO: Visio-tactile Implicit Representations of Deformable Objects., Youngsun Wi, Pete Florence, Andy Zeng, Nima Fazeli. ICRA 2022.
>
> A1: We thank the reviewer for bringing these recent works to our attention! We have added references to the mentioned works in the related works section:
>
> &nbsp;
>
>
> The main text, section 2.1:
>
> > *The progress in Deformable Object Manipulation (DOM) partially stems from the recent emergence of DOM simulators in the past years. These benchmarks and simulators have significantly advanced the state-of-the-art in modeling the complex dynamics of DOM and provide the firmament of the algorithmic innovations for the DOM methods. SoftGym \citep{softgym} is the first DOM benchmark that models all liquid, fabric, and elastoplastic objects and introduces a wide range of standardized DOM tasks.* ***ThreeDWorld \citep{TDW} allows agents to physically interact with various objects in rich 3D environments. However, they are non-differentiable and thus do not support methods based on differentiable physics.***
>
> &nbsp;
>
> The main text, section 2.2, Motion Planning:
>
> > ***\textbf{Motion planning} for DOM has been explored in \citep{ACID, VIRDO, VFMPlanning, HF_planning, CE_planning}.*** *These methods overcome the enormous DoFs of the original state space by learning a low-dimensional latent state representation and a corresponding dynamic model for planning. The success of these planning methods critically depends on the dimensionality and quality of the learned latent space and the dynamic model, which in itself is challenging.*
>
> &nbsp;
>
> > **Q2**: Saying the simulator has sim2real ability is a pretty big claim. However, only a demo video is provided without any systematic evaluation.
>
> A2: Thank you for raising this concern. We concur with the reviewer that our hardware experiments only support reasonable fidelity.
>
> Having said that, prior works show that if the simulator is differentiable, the sim-to-real gap can be minimized by system identification [1, 2]. We envision that the sim-to-real issues can be mitigated by taking advantage of DaX’s differentiability. Due to the limited scope of the paper, providing interfaces for users to do so in DaXBench and DaX can be our future work. Therefore, we have modified the descriptions and removed the overly strong claims in the updated manuscript:
>
> &nbsp;
>
> > ***Granted that the physical engine of DaX is built upon state-of-the-art simulation methods, the fidelity and correctness still need to be tested. To testify to the correctness of our simulated dynamics to real-world physics, we carry out a real robot experiment on a Kinova Gen 3 robot.*** *We deploy CEM-MPC with DaX as the predictor model to a Push-Rope task, as shown in Figure 5. Our study finds that the resultant state trajectories are similar in simulation and in reality.* ***This validates the fidelity of the tasks implemented based on our engine, and the performance of the DOM methods in our simulator provides hints about their performance on the real robot. We refer the reader to the appendix for details. In addition, the differentiability of DaX enables system identification that can reduce the sim-2-real gap (Le Lidec et al., 2021; Murthy et al., 2021). We can get an error term by comparing the observations in simulation and in reality; by differentiating this error with respect to the simulator parameters, we get gradients for parameter tuning.***
>
> &nbsp;
>
> [1] Murthy, J. Krishna, et al. "gradSim: Differentiable simulation for system identification and visuomotor control." International Conference on Learning Representations. 2020.
>
> [2] Le Lidec, Quentin, et al. "Differentiable simulation for physical system identification." IEEE Robotics and Automation Letters 6.2 (2021): 3413-3420.

---

### Official Review · Reviewer_4Kpq · 2022-10-23

**Confidence:** 3
**Correctness:** 3
**Technical Novelty And Significance:** 3
**Empirical Novelty And Significance:** 3
**Recommendation:** 8

**Clarity, Quality, Novelty And Reproducibility:**

The Quality of this paper is quite good to me as it covers different algorithms, tasks, and physics systems. The analysis after the experiments is insightful and valuable for future research.

The novelty of this paper is okay as it designs an efficient differentiable framework to simulate soft bodies. It could be better if it can provide more quantitative evaluations of the performance (e.g. comparison with PlasticineLab)

The experiments and methods look reasonable. And the authors promise to release the entire benchmark and the code, I think this paper should be reproducible.

The clarity is fine. Here are some sentences that I might need more clarification on,

‘The policy complexity grows exponentially with the planning horizon, known as “curse of dimensionality”;’ - This is not how I usually understand the “curse of dimensionality”. Are there more concrete citations or examples of the exponential complexity w.r.t. the horizon?

‘Macro-actions reduce the dimension of the action space and the effective horizon, and hence increase the scalability requirement on the DOM methods’ - Why does it increase the requirements instead of making the scaling easier?


**Strength And Weaknesses:**

Strength

In my view, the experiments are the highlights of this paper. The authors implement the recently proposed differentiable simulation papers in their benchmark DaX, and compare them with gradient-free methods. The experiments include various robotic learning algorithms. This systematic evaluation indicates that the differentiable-physics-based method can indeed have better performance in some cases, but they also suffer from limitations as many gradient-based methods do, i.e. sensitive to the optimization landscape and initialization. And it also reiterates the importance of exploration in those differentiable-physics-based methods. Moreover, the authors perform real-world experiments, making the benchmark more convincing.

Weaknesses

Since the research of differentiable physics is progressing fast, the authors might better illustrate and clarify its difference from related literature. For example, PlasticineLab is another platform for differentiable soft body manipulation. Since both DaX (fluid, elastoplastic, and mixture) and PlasticineLab are using MPM as the governing dynamics, it is not apparent to me why PlasticineLab cannot support liquid and mixture (in Table 1) while DaX can. Moreover, it is unclear to me how the solid ingredients are represented in Pour-Soup. Are they elastoplastic objects or purely rigid bodies? Is there a coupling issue between the fluid particles and the ingredients?
Furthermore, the Saving Memory strategy in Section 3.1 is the same as the checkpoint scheme in [1] (Section 4.2), where the back-propagation trades time for memory usage by recomputing the intermediate variables in each step.

[1] Qiao, Yi-Ling, Junbang Liang, Vladlen Koltun, and Ming C. Lin. "Efficient differentiable simulation of articulated bodies." In International Conference on Machine Learning, pp. 8661-8671. PMLR, 2021.


**Summary Of The Paper:**

This paper presents a benchmark that can simulate and evaluate a variety of manipulation tasks on deformable objects, including fluid, cloth, rope, and elastoplastic objects. The simulator is implemented using Jax, therefore it can also provide analytic gradients, making it compatible with recent popular control algorithms designed for differentiable simulation.

A series of comparisons between differentiable-simulation-physics methods and gradient-free ones are further conducted. The experiments cover a wide range of categories of algorithms, e.g. reinforcement learning, imitation learning, and planning. Based on these comprehensive evaluations, the authors conclude the limitation and advantages of the differentiable-physics-based methods.

In the end, the authors successfully conduct a real-world Push-Rope task using the control policy from the simulation, which can validate the fidelity of the simulator.


**Summary Of The Review:**

In my view, this paper provides a unified platform to evaluate the soft-body manipulation tasks and differentiable physics algorithms. It conducts comprehensive experiments and provides useful insights for future work. The sim-to-real experiment makes it more convincing. One thing it can improve is its differentiation from previous methods. In total, I think it is a good paper.

---

> ### Author Response · Authors · 2022-11-14
> **Authors Response**
>
>
> > **Q3**:  Furthermore, the Saving Memory strategy in Section 3.1 is the same as the checkpoint scheme in [1].
>
> A3: Thank you for bringing this paper to our attention! We have checked the “checkpointing method” in section 4.2 and verified both methods use the same idea. We have added the references to the prior works on the “checkpointing method” and adjusted our manuscript as follows:
>
> &nbsp;
>
> The main text, Section 3.1, Saving memory:
>
> > *Saving Memory Dax uses two tricks to optimize the efficiency and memory consumption of the end-to-end gradient computation, namely lazy dynamic update and ***''checkpointing method'' [1, 2, 3]***. To optimize MPM's time and memory consumption for a single timestep, Dax lazily updates the dynamic of a small region affected by the manipulation at that step. In addition, Dax only stores the values of a few states **(i.e. checkpoints)** at the forward pass; during the backward propagation, Dax re-computes segments of the forward values starting from every stored state in the reverse order sequentially for the overall gradient. We refer the reader to the appendix for details.*
>
> &nbsp;
>
> Appendix A.4, Checkpointing method:
>
>
> > ***Checkpointing method  Dax only stores the values of a few states (i.e. checkpoints)***  *and re-evaluates the sub-step gradients online during the back-propagation to reduce the memory consumption for long horizon problems. ***The checkpointing method has been proven useful in saving memory consumption for differentiable simulations in many prior works [1, 2, 3].*** … To overcome the large memory consumption issue, Dax trade-offs the computation complexity slightly for large memory consumption improvement: Dax only stores the values of a few states ***(i.e. checkpoints)*** at the forward pass; during the backward propagation, Dax re-computes segments of the forward values starting from every stored state in the reverse order sequentially for the overall gradient.*
>
> &nbsp;
>
> References
>
> [1] Yi-Ling Qiao, Junbang Liang, Vladlen Koltun, and Ming C. Lin. Efficient differentiable simulation of articulated bodies. In ICML, 2021.
>
> [2] Andreas Griewank and Andrea Walther. Algorithm 799: Revolve: An implementation of checkpointing for the reverse or adjoint mode of computational differentiation. ACM Trans. Math. Softw., 26(1):19–45, Mar 2000. ISSN 0098-3500.
>
> [3]  Tianqi Chen, Bing Xu, Chiyuan Zhang, and Carlos Guestrin. Training deep nets with sublinear memory cost, 2016. URL https://arxiv.org/abs/1604.06174.
>
> &nbsp;
>
> > **Q4**:  ‘The policy complexity grows exponentially with the planning horizon, known as “curse of dimensionality”;’ - This is not how I usually understand the “curse of dimensionality”. Are there more concrete citations or examples of the exponential complexity w.r.t. the horizon?
>
> A4: This is an excellent point for us to clarify! We meant that the planning methods' time complexity grows exponentially with the planning horizon [1, 2]. The dependency between the policy complexity and the planning horizon has been studied in [3], proving that the policy complexity grows non-trivially with the planning horizon. More specifically, it shows that the planning horizon monotonically adjusts the size of the policy class from 1 to at least $|A|^{|S|−2}$, which is "almost all" of the $|A|^{|S|}$ possible policies. Nevertheless, such growth may not be exponential with the planning horizon. We apologize for the inaccurate description in the script, and we have revised the manuscript as follows:
>
> &nbsp;
>
> > *The ***time complexity for the search-based planning method grows exponentially with the planning horizon, known as the ''curse of dimensionality''***; this is exacerbated by the large state space of deformable objects.*
>
> &nbsp;
>
> [1] Russell, Stuart and Norvig, Peter (2010). "Artificial Intelligence: A Modern Approach". Prentice Hall.
>
> [2] Silver, David, and Joel Veness. "Monte-Carlo planning in large POMDPs." Advances in neural information processing systems 23 (2010).
>
> [3] Nan Jiang, Alex Kulesza, Satinder P. Singh, Richard L. Lewis, The Dependence of Effective Planning Horizon on Model Accuracy, AAMAS, 2015.
>
> &nbsp;
>
> > **Q5**:  Why do macro actions increase the requirements instead of making the scaling easier?
>
> A5: Thank you for pointing out the misused “requirements.” Indeed, macro-actions make scaling easier. We have fixed and refined the description in the updated manuscript:
>
> *Macro-actions reduce the dimension of the action space and the effective horizon, and hence make ***scaling easier*** for the DOM methods.*

---

> ### Author Response · Authors · 2022-11-14
> **Authors Response**
>
> Thank you for acknowledging our effort and the contribution of DaxBench! We would like to first compare the difference between DaxBench and the existing related literature, then clarify some other issues on the paper’s presentation. We’ve updated the manuscript in response to some of your suggestions. Thank you for helping us in making this work better!
>
> &nbsp;
>
> > **Q1**: The authors might better illustrate and clarify its difference from related literature. For example, PlasticineLab is another platform for differentiable soft body manipulation. It is not apparent to me why PlasticineLab cannot support liquid and mixture (in Table 1) while DaX can.
>
> A1: PlasticineLab uses the MLS-MPM method that can support both liquids and mixtures. However, the simulator PlasticineLab focuses only on elastic-plastic tasks and does not implement liquids and mixtures. Therefore, it is non-trivial to extend PlasticineLab to support mixtures of different materials. DaXBench implements these tasks to facilitate researchers working with deformable objects using JAX.
>
>
> &nbsp;
>
> > **Q2**. Moreover, it is unclear to me how the solid ingredients are represented in Pour-Soup. Are they elastoplastic objects or purely rigid bodies? Is there a coupling issue between the fluid particles and the ingredients?
>
> A2: The ingredients, including tofu and broccoli, in the Pour-Soup task, are modeled as elastic objects using particles. The mixture is modeled using the MLS-MPM method [1], a two-way coupling method. The MLS-MPM method naturally handles the collisions of mixtures through particle-to-grid (P2G), grid operation (GO), and then grid-to-particle (G2P) processes.
>
> &nbsp;
>
> [1] Hu, Y., Fang, Y., Ge, Z., Qu, Z., Zhu, Y., Pradhana, A., & Jiang, C. (2018). A moving least squares material point method with displacement discontinuity and two-way rigid body coupling. ACM Transactions on Graphics (TOG), 37(4), 1-14. [SIGGRAPH18]

---

### Official Review · Reviewer_yxby · 2022-10-25

**Confidence:** 3
**Correctness:** 3
**Technical Novelty And Significance:** 3
**Empirical Novelty And Significance:** 3
**Recommendation:** 8

**Clarity, Quality, Novelty And Reproducibility:**

The manuscript is nicely written. I did not follow the *Lazy Dynamic Update* part and hope for more explanation of it.

**Strength And Weaknesses:**

Strength:
- As far as I know, DaXBench is the first deformable object manipulation benchmark platform based on JAX.
- This manuscript compared a large set of methods on their proposed method, which provides the readers with a good sense of the performance of their simulator and the compared methods.
- The provided code represents the visible reproducibility of the work.

Weaknesses:
- I wonder how the *mixture* is handled in these demos: are they two-way coupled, one-way coupled, or any other coupling was used?
- I personally did not follow the *Lazy Dynamic Update* part: MPM will require a grid for simulation. Does it mean the overall grid is 32*6*32, or is the grid actually 128*128*128, but only a small region of it is activated? My personal feeling is this setting only applies to a small fraction of the proposed environments, is it?
- My main reservation is the comparison at the framework level. JAX has a very similar position to taichi and Nvidia warp, which are more famous for physical simulation. And I would expect they have similar performance to JAX-implemented environments, is it? Is DaXBench distinguishable from them because of user-friendliness? Or better interface to deep learning modules? Or speed?

**Summary Of The Paper:**

The manuscript proposed a benchmark platform for deformable object manipulation tasks: DaXBench. DaxBench contains several environments with a mixture of fluid, fabric, and elastoplastic materials. Due to the support of JAX, DaXBench seems to support multi-GPU simulation and training easily. Also, it promotes differentiable simulation. The authors compared a set of controller optimization methods on the proposed benchmark platform and shared their insights.

**Summary Of The Review:**

This manuscript proposed DaxBench, the first deformable object manipulation benchmark platform based on JAX. I appreciate the work from the authors to build the simulation framework and share it with the community, but I expect some explanations and comparisons to taichi/warp.

---

> ### Author Response · Authors · 2022-11-14
> **Authors Response**
>
> > **Q3**: Is DaXBench distinguishable from Tachi, and Nvidia warp, because of user-friendliness? Or better interface to deep learning modules? Or speed?
>
> A3: We expect JAX, NVIDIA Warp, and Taichi to have similar computational speeds.
>
> We distinguish DaxBench from other simulators based on Taichi or NVIDIA Warp regarding user-friendliness and integration to deep learning frameworks for robot learning.
> JAX is user-friendly, e.g., it has the same interface as NumPy and offers a full tensor-based programming model.
> JAX is well integrated with deep learning frameworks, e.g., Flax [1] is a neural network framework implemented in JAX, and RLax [2] is a library that exposes valuable building blocks for implementing reinforcement learning agents. Both libraries are well supported by large research organizations like Google Brain and DeepMind.
>
> Taking differentiable RL, for example, both the simulator and the RL algorithm can be written in JAX. Hence, the gradient can directly propagate from the simulated physical quantities to the parameters of policy and value functions of the RL algorithm. In contrast, other simulators written in Taichi and Warp has to implement their RL methods in other software libraries, such as PlasticineLab [3] with PyTorch; as a result, during learning, the gradients need to be converted from the simulation framework to the learning framework, which typically causes computational overhead.
>
>
>
> [1] Jonathan Heek, Anselm Levskaya, Avital Oliver, Marvin Ritter, Bertrand Rondepierre, Andreas Steiner, Marc van Zee, Flax: A neural network library and ecosystem for JAX, http://github.com/google/flax.
>
> [2] Babuschkin, Igor and Baumli, Kate and Bell, Alison and Bhupatiraju, Surya and Bruce, Jake and Buchlovsky, Peter and Budden, David and Cai, Trevor and Clark, Aidan and Danihelka, Ivo and Fantacci, Claudio and Godwin, Jonathan and Jones, Chris and Hemsley, Ross and Hennigan, Tom and Hessel, Matteo and Hou, Shaobo and Kapturowski, Steven and Keck, Thomas and Kemaev, Iurii and King, Michael and Kunesch, Markus and Martens, Lena and Merzic, Hamza and Mikulik, Vladimir and Norman, Tamara and Quan, John and Papamakarios, George and Ring, Roman and Ruiz, Francisco and Sanchez, Alvaro and Schneider, Rosalia and Sezener, Eren and Spencer, Stephen and Srinivasan, Srivatsan and Wang, Luyu and Stokowiec, Wojciech and Viola, Fabio, The Deep Mind JAX Ecosystem, http://github.com/deepmind.
>
> [3] Zhiao Huang, Yuanming Hu, Tao Du, Siyuan Zhou, Hao Su, Joshua B. Tenenbaum, Chuang Gan, PlasticineLab: A Soft-Body Manipulation Benchmark with Differentiable Physics, International Conference on Learning Representations, 2021.

---

> ### Author Response · Authors · 2022-11-14
> **Authors Response**
>
> Thank you for recognizing the contribution of our work! You’ve raised a few great points and helped us to  further polish our manuscript. We would like to address your questions/concerns in the following order.
>
> &nbsp;
>
> > **Q1**: I wonder how the mixture is handled in these demos: are they two-way coupled, one-way coupled, or any other coupling was used?
>
> A1: The mixture handled in the demonstration is two-way coupled. It is implemented based on the MLS-MPM method [1], a two-way coupling method. In the soup simulation, all gradients, including tofu, broccoli, and soup, are simulated with particles. MLS-MPM naturally handles the collisions of mixtures through particle-to-grid (P2G), grid operation (GO), and then grid-to-particle (G2P) processes.
>
>
> [1] Hu, Y., Fang, Y., Ge, Z., Qu, Z., Zhu, Y., Pradhana, A., & Jiang, C. (2018). A moving least squares material point method with displacement discontinuity and two-way rigid body coupling. ACM Transactions on Graphics (TOG), 37(4), 1-14. [SIGGRAPH 18]
>
> &nbsp;
>
> > **Q2**: I personally did not follow the Lazy Dynamic Update part, clarification is needed.
>
> A2: This is an excellent point for us to elaborate more on! The overall grid size is generally large and fixed for each task, while the size of the active region, for which region we update the dynamic, is much smaller and adjustable. Taking the Rope-Push task, for example, the size of the overall grid is 128 $\times$ 128 $\times$ 128, it only a small region of size 32 $\times$  6 $\times$ 32 is activated for the dynamic update.
>
> Intuitively, this trick is derived from the empirical observation that for many tasks, only a small region needs to update its dynamics for each time step. In addition, for those tasks in which the objects can potentially occupy the entire space, we observe that only a small region’s dynamic is required to solve the tasks. Taking the PourWater task for example, though the water can be spilled to cover the entire space, to pour the water successfully, our simulator only needs to accurately model the water when it is still in the bowl. The Lazy Dynamic Update trick can be applied to both of these scenarios.
>
> The effectiveness of this trick is indeed task-dependent. Admittedly, this trick only partially resolves the memory issue since we can easily construct a task for which the deformable object occupies the entire grid, which necessitates the active region to cover the entire grid. However, this trick does help all the current task environments in DaxBench to various extents. We provide a table below to summarize and contrast the sizes of the original grid and its corresponding active region for each task environment. The last column shows that the Lazy Dynamic Update has reduced the state size by more than 75% for all environments.
>
> &nbsp;
>
> |Task|Activation size|Full-size|Volume Reduction|
> | - | - | - | - |
> |Fabric Manipulation Tasks|N.A.|N.A.|N.A.|
> |Rope|Shape rope: 32 $\times$  6 $\times$  32|128 $\times$  128 $\times$  128|99.7%|
> ||Wipe rope: 32 $\times$  32 $\times$  32|64 $\times$  64 $\times$  64|87.5%|
> |Water|40 $\times$  40 $\times$  40|80 $\times$  80 $\times$  80|87.5%|
> |Mixture|40 $\times$ 40 $\times$ 40|80 $\times$ 80 $\times$ 80|87.5%|

---

### Decision · Program_Chairs · 2023-01-20

**Decision:**

Accept: notable-top-5%

**Justification For Why Not Higher Score:**

Already highest level of score.

**Justification For Why Not Lower Score:**

The meta-reviewer shares the same excitement as all other reviewers that this work is an important contribution to the community.

**Metareview: Summary, Strengths And Weaknesses:**

This paper proposes a benchmark for deformable object manipulation, which is based on differentiable physics and uses the latest JAX framework. The support by JAX and differentiable property will benefit many algorithms in this direction. All reviewers are positive about the contribution. Some issues regarding the details are raised by the reviewers and then addressed by the authors. It is a clear case of an accepted research work at ICLR.

**Note From Pc:**

if the above contains the word "oral" or "spotlight" please see: "oral" presentation means -> notable-top-5% and "spotlight" means -> notable-top-25%. As stated in our emails, we are disassociating presentation type from AC recommendations

**Summary Of Ac-Reviewer Meeting:**

Not a borderline paper.